# Perinatal exposure to a human relevant mixture of persistent organic pollutants: Effects on mammary gland development, ovarian folliculogenesis and liver in CD-1 mice

Silje Modahl Johanson[1]*, Erik Ropstad[1], Gunn Charlotte Østby[1], Mona Aleksandersen[2], Galia Zamaratskaia[3], Gudrun Seeberg Boge[4¤], Ruth Halsne[5], Cathrine Trangerud[4], Jan Ludvig Lyche[6], Hanne Friis Berntsen[1,7], Karin Elisabeth Zimmer[2], Steven Verhaegen[1]

1 Department of Production Animal Clinical Sciences, Norwegian University of Life Sciences, Ås, Norway,
2 Department of Preclinical Sciences and Pathology, Norwegian University of Life Sciences, Ås, Norway,
3 Department of Molecular Sciences, Swedish University of Agricultural Sciences, Uppsala, Sweden,
4 Department of Companion Animal Clinical Sciences, Norwegian University of Life Sciences, Ås, Norway,
5 Division of Laboratory Medicine, Department of Forensic Sciences, Oslo University Hospital, Oslo, Norway,
6 Department of Paraclinical Sciences, Norwegian University of Life Sciences, Ås, Norway, 7 National Institute of Occupational Health, Oslo, Norway

¤ Current address: The Norwegian Medicines Agency, Oslo, Norway
* Silje.modahl.johanson@nmbu.no

## Abstract

The ability of persistent organic pollutants (POPs) with endocrine disrupting properties to interfere with the developing reproductive system is of increasing concern. POPs are transferred from dams to offspring and the high sensitivity of neonates to endocrine disturbances may be caused by underdeveloped systems of metabolism and excretion. The present study aimed to characterize the effect of *in utero* and lactational exposure to a human relevant mixture of POPs on the female mammary gland, ovarian folliculogenesis and liver function in CD-1 offspring mice. Dams were exposed to the mixture through the diet at Control, Low or High doses (representing 0x, 5000x and 100 000x human estimated daily intake levels, respectively) from weaning and throughout mating, gestation, and lactation. Perinatally exposed female offspring exhibited altered mammary gland development and a suppressed ovarian follicle maturation. Increased hepatic cytochrome P450 enzymatic activities indirectly indicated activation of nuclear receptors and potential generation of reactive products. Hepatocellular hypertrophy was observed from weaning until 30 weeks of age and could potentially lead to hepatotoxicity. Further studies should investigate the effects of human relevant mixtures of POPs on several hormones combined with female reproductive ability and liver function.

## Introduction

Persistent organic pollutants (POPs) are of significant concern due to their high resistance to degradation and potential toxicity [1]. This has led to the regulation of production, use and

**Data Availability Statement:** All raw data are available from the DataverseNO database at DOI: https://doi.org/10.18710/MSTY0R.

**Funding:** The present study was funded by the Norwegian Research Council, Grant numbers 21307 and 204361 (ER). The funders played no role in the study design, data collection and analysis, decision to publish, or preparation of the manuscript.

**Competing interests:** The authors have declared that no competing interests exist.

release of POPs by the Stockholm Convention [2]. A variety of POPs exhibit endocrine disrupting properties as they interfere with the synthesis, transport, metabolism, or elimination of endogenous hormones. The hormonal system is of vital importance in the fully functioning organism. Thus, there is a growing concern about endocrine disrupting chemicals (EDCs) and their interference with the reproductive, metabolic, neuroendocrine or cardiovascular systems [3, 4]. POPs are known to be transferred from mothers to offspring through the placenta and breast milk [5–8]. As early stages of development are particularly sensitive to disturbances by EDCs, the maternal transfer of POPs may have devastating consequences [9–11].

Mammary gland (MG) development is an intricate process finely regulated by hormones, growth factors and stromal factors, all of which can be disturbed by EDCs [12, 13]. One extensively studied example is the effect of perfluorooctanoic acid (PFOA) on MG development in rodents [14–19]. Other studies have also reported effects in the developing female MG or alterations in ovarian follicle maturation by various EDCs (extensively reviewed in [3] and [4]). Changes in ovarian folliculogenesis can lead to premature ovarian insufficiency and infertility [20, 21]. Furthermore, exposure to EDCs may increase the risk of developing breast, cervical, uterine or ovarian cancer [22].

The high sensitivity of fetuses and neonates to POP exposure, compared to adults, can be partly explained by immature metabolic and excretion systems [23, 24]. The cytochrome P450 (CYP) family of enzymes (included in phase I metabolism) catalyzes the transformation of xenobiotics to more polar derivatives that can be further metabolized or excreted [25–27]. However, transformation by CYP enzymes may also form free radicals (e.g. reactive oxygen species) or activate procarcinogens, causing potentially more deleterious effects than the original compound [28, 29]. The expression of CYP enzymes is regulated by different nuclear receptors such as the aryl hydrocarbon receptor (AHR), the pregnane X receptor (PXR) and the constitutive androstane receptor (CAR) [30, 31]. Various POPs have been shown to stimulate these receptors and, consequently, increase the enzymatic activities of CYPs in rodents [32–35]. Furthermore, upregulation of hepatic CYP transcripts at early life stages has been stated as early indicators of liver toxicity and carcinogenicity [36].

Previously, toxicological studies focused on the use of single compounds or technical mixtures of POPs. However, recent attention has shifted to study more realistic exposure scenarios with multiple compounds that may interact, and cause effects not anticipated by dose addition [37, 38]. For the present study, a complex mixture of POPs was designed based on information from Scandinavian food basket surveys [39]. The mixture included polychlorinated biphenyls (PCBs), organochlorine pesticides (OCPs), brominated flame retardants (BFRs) and perfluoroalkylated substances (PFASs). The present study aimed to investigate how *in utero* and lactational exposure to the mixture of POPs affected the developing female reproductive organs (the MG and ovary) and liver function in CD-1 mice.

## Materials and methods

### Ethical considerations

The study was approved by the Institutional Animal Care and Use Committee at the Norwegian University of Life Sciences (NMBU) and the Norwegian Food Safety Authority (application ID: FOTS 7722). It was conducted in accordance with The Norwegian Regulation on Animal Experimentation at the Section for Experimental Biomedicine, NMBU-Faculty of Veterinary Medicine, in Oslo, Norway. The animals were health-monitored according to recommendations by the Federation of European Laboratory Animal Science Association (FELASA; http://www.felasa.eu/) and kept under Specific Pathogen Free (SPF) conditions.

## Feed design

The POP mixture was designed by Berntsen and colleagues [39]. In brief, the composition of PCBs, OCPs, BFRs and PFASs was chosen based on concentrations in Scandinavian food products stated in publications prior to 2012. Human estimated daily intake (hEDI) levels were estimated and adapted to a 25 g mouse consuming 3 g feed/d. The concentrations were upward adjusted to 5000x (Low dose) and 100 000x (High dose) hEDI. All polybrominated diphenyl ethers (PBDEs), PCBs and OCPs were purchased from Chiron AS (Trondheim, Norway). Hexabromocyclododecane (HBCD) and all PFASs, except for perfluorohexane sulfonic acid (PFHxS) purchased from Santa Cruz Biotechnology Inc. (Dallas, USA), were obtained from Sigma-Aldrich (St. Louis, USA). Compounds were dissolved in acetone, cyclohexane or chloroform and added to corn oil (Jasmin, fully refined, Yonca Gida San A.S., Manisa, Turkey). The solvent was evaporated under $N_2$-flow and the mixture was incorporated into AIN-93G mouse feed (TestDiets, St.Louis, USA). The control diet contained corn oil from which the solvent had been evaporated. An additional AIN-93G reference diet was also made using corn oil instead of soybean oil. Mixture composition and concentrations of individual compounds are presented in Table 1.

## Animals, experimental design, and sample collection

Timed-pregnant CD-1 mice (F0) were obtained from Charles River Laboratories (Wilmington, USA) and gave birth 4–5 d after arrival to the F1 generation. At 3 weeks of age, F1 females were randomly assigned to a dose: Control (n = 28), Low (n = 27), or High (n = 20). Exposure continued throughout mating (at 10 weeks of age with CD-1 males bred in-house), gestation and lactation. Offspring (F2) produced by the F1 dams were given the AIN-93G reference diet from 3 weeks of age and only exposed to the mixture of POPs *in utero*, through lactation and by nibbling on their mothers' feed prior to weaning. The study design in illustrated in S1 Fig.

All mice were housed in closed Type III individually ventilated cages (IVC) (Allentown Inc, USA) except during mating when animals were moved to open Makrolon Type III cages (Techniplast, Buguggiate, Italy). All cages contained standard aspen bedding, red polycarbonate houses and cellulose nesting material (Scanbur BK, Karlslunde, Denmark). Water and feed were available *ad libitum*. Cages, bedding, nesting material and water bottles were changed weekly. The animal room was on a 12:12 light-dark cycle, with a room temperature of 21 ± 2°C and 45 ± 5% relative humidity.

Dams (F1) were euthanized at gestation d 17 (pregnant; n = 12, 16 and 8 for the Control, Low and High doses, respectively) or at 21 d post-partum (post-pregnant; n = 14, 10 and 11 for the Control, Low and High doses, respectively) with a total exposure time of 9 or 13 weeks, respectively. Mating was unsuccessful for 3 dams (2 in Control and 1 in High). One dam (Low dose) died prior to mating.

Female offspring (F2) were sacrificed at 3 (weaning), 6 (pubertal) and 9 (adult) weeks of age (n = 12, 14 and 14 for all doses, respectively). Male offspring (F2) were sacrificed at 9 or 30 weeks of age (n = 15 for all doses) to further investigate the hepatic alternations observed in females. Due to a low number of males produced by dams exposed to the High dose, all males in this dose were sacrificed at 9 weeks of age. Thus, 30-week-old male offspring were only sampled from the Control and Low doses.

Mice were sacrificed under anesthesia (isoflurane gas obtained from Baxter, San Juan, Puerto Rico) by cardiac puncture and cervical dislocation. The 4th MG (left), the ovaries and a section of the liver were fixed in 10% neutral buffered formalin for histopathological examinations. The remaining liver was frozen on dry ice and stored at -80°C for analysis of chemicals

**Table 1. POP concentrations in feed.**

| Compounds | Nominal concentrations Low | Measured concentrations Low | Nominal concentrations High | Measured concentrations High |
|---|---|---|---|---|
| **Polychlorinated biphenyls (PCBs)** | | | | |
| PCB-28 | 5.8 | 3.1 | 117 | 46 |
| PCB-52 | 13.8 | 15.0 | 275 | 182 |
| PCB-101 | 23.3 | 25.4 | 467 | 377 |
| PCB-118 | 40.4 | 37.2 | 808 | 612 |
| PCB-138 | 57.5 | 53.8 | 1150 | 957 |
| PCB-153 | 57.5 | 61.4 | 1150 | 981 |
| PCB-180 | 15.4 | 17.4 | 308 | 263 |
| Σ7PCBs | 213.7 | 213.3 | 4275 | 3418 |
| **Organochlorine pesticides (OCPs)** | | | | |
| HCB | 50.0 | 37.4 | 1000 | 588 |
| α-chlordane | 37.5 | 45.0 | 750 | 723 |
| Oxychlordane | 12.5 | 9.8 | 250 | 297 |
| *Trans*-nonachlor | 12.5 | 14.9 | 250 | 264 |
| α-HCH | 21.7 | 21.2 | 433 | 421 |
| β-HCH | 17.5 | 22.3 | 350 | 398 |
| γ-HCH (Lindane) | 23.8 | 31.4 | 475 | 435 |
| Σ7OCPs | 175.5 | 182.0 | 3508 | 3126 |
| *p,p'*-DDE | 119.6 | 136.0 | 2392 | 2390 |
| Dieldrin | 75.0 | 70.4 | 1500 | 1470 |
| **Brominated flame retardants (BFRs)** | | | | |
| BDE-47 | 40.4 | 39.7 | 808 | 642 |
| BDE-99 | 7.9 | 8.6 | 158 | 126 |
| BDE-100 | 6.3 | 5.6 | 125 | 91 |
| BDE-153 | 1.3 | 1.5 | 25 | 22 |
| BDE-154 | 2.5 | 2.8 | 50 | 38 |
| ΣBDE-47-154 | 58.4 | 58.2 | 1166 | 919 |
| BDE-209 | 62.5 | 64.8 | 1250 | 1141 |
| HBCD | 12.5 | 9.9 | 250 | 203 |
| **Perfluoroalkylated substances (PFASs)** | | | | |
| PFHxS | 4.9 | 1.7 | 98 | 42 |
| PFOS | 10.8 | 3.2 | 217 | 74 |
| PFOA | 18.3 | 6.0 | 367 | 121 |
| PFNA | 5.8 | 2.1 | 117 | 42 |
| PFDA | 7.9 | 3.1 | 158 | 57 |
| PFUnDA | 4.0 | 1.6 | 80 | 28 |
| ΣPFASs | 51.7 | 17.7 | 1037 | 364 |

Nominal and measured concentrations (ng/g feed) of the PCBs, OCPs, BFRs and PFASs in AIN-93G mouse feed at the Low and High doses (5000x and 100 000x human estimated daily intake, respectively). Adapted from Table 1 in [39].

and cytochrome P450 activity. The 4[th] MG (right) was peeled away from the inner skin surface and spread out on a microscope slide for whole mount analysis.

## Analysis of mammary glands

**Whole mount preparation, development scoring and Sholl analysis.** Whole mounts were used to assess alterations in MG development of female offspring (F2). The 4[th] MG from the right side was placed with the skin side down on a charged microscope slide and fixed in

Carnoy's solution (6:3:1 of ethanol:chloroform:acetic acid) at 4˚C overnight. Whole mounts were rinsed in 70% ethanol, gradually transitioned to water and stained overnight in carmine alum stain (2 g/L carmine and 5 g/L aluminum potassium sulfate). After staining, the whole mounts were dehydrated gradually to 100% ethanol and de-fatted in xylene overnight until visibly clear. Multiple pictures were taken of each whole mount at 10x magnification using a Zeiss Axio Imager light microscope and camera (M2, Oberkochen, Germany), and a mosaic [40] was created using ImageJ [41].

Qualitative development scoring was conducted on a scale from 1 to 4 (1 = poor development, 4 = best development). The following criteria were evaluated: lateral and longitudinal epithelial growth into the surrounding fat pad; branching degree; alveolar budding; lobule formation; and the presence or absence of terminal end buds (TEBs). Scores were assigned by two assessors and repeated 3x for each whole mount to give a median score from each assessor. The final score was obtained by averaging the two median scores. The assessors were initially not blinded to Control as to establish a baseline of MG development. Subsequently, the glands were evaluated blinded to dose within each age class. TEBs were defined as ends with a diameter of $\geq$ 100 μm and counted using ImageJ.

MG branching density was quantitatively assessed using ImageJ and the modified Sholl analysis [42]. In brief, the color channels were separated, and noise were removed from the images using various methods supplied by ImageJ. The images were skeletonized, made binary and dilated 1x to fill in gaps created by skeletonization. Longitudinal distance of the gland (termed mammary epithelial length) was measured between the base of the epithelial tree and the tip of the most distal branch on the skeletonized images. The mammary epithelial area (MEA) and the lymph node area (LNA) were defined by the periphery of the total skeletonized gland and the area covered by the lymph node, respectively. The total number of intersections (N) within the MEA was determined by Sholl analysis. Branching density was calculated by N/(MEA-LNA). A detailed description of the method is given by Stanko and Fenton [43].

Two whole mount images (6 weeks Control) were ruined during processing. In addition, three glands were partly torn during whole mount preparation (1 from 3 weeks High, 1 from 9 weeks High and 1 from 9 weeks Control). Thus, a total of 5 whole mounts were excluded from the Sholl analysis.

**Image analysis of glandular tissue.** The 4[th] MG from the left side of female offspring (F2) were fixated in formalin for at least 24 h. Glands were transferred to 70% ethanol prior to paraffin embedding and sectioned horizontally using a microtome (5 μm). Gland orientation for sectioning was ensured identical to whole mount orientation. Sections were attached onto glass slides and stained with haematoxylin and eosin (HE). Sections were examined blindly by a board-certified pathologist.

Digital photomicrographs of the glands were taken at 100x magnification using Zeiss Axio Imager light microscope, Axiocam 506 color camera and Zen microscope software (Carl Zeiss Microscopy GmbH, Germany). Photomicrographs were stitched and reconstructed digital images of the whole glands were created. Image-Pro Plus (Media Cybernetics Inc., MD, USA) was used to measure area fractions occupied by glandular tissue.

## Analysis of ovaries

Both ovaries from female offspring (F2) were fixated in 10% neutral buffered formalin (for 24 h), transferred to 70% ethanol and embedded in paraffin. Sections (8 μm) were made using a microtome, every 10[th] section was mounted onto glass slides and stained with HE. Histological examination was conducted using a light microscope on every 2[nd] glass slide throughout one randomly chosen ovary. Between 9 and 12 females were randomly selected for evaluation from

each dose and sampling time. One observer, blinded to treatment, counted the numbers of healthy follicles, including primordial follicles, primary follicles, preantral follicles and antral follicles, as categorized by Flaws and colleagues [44]. Primordial follicles were defined as having a single layer of squamous granulosa cells surrounding an oocyte, while primary follicles consisted of an oocyte surrounded by one layer of cuboidal granulosa cells. Preantral follicles were defined as being surrounded by at least two layers of cuboidal granulosa cells and theca cells, and antral follicles consisted of a fluid-filled antral space adjacent to the oocyte and surrounded by many layers of theca cells and cuboidal granulosa cells. The preantral and antral follicles were only registered if the oocyte showed nuclear material. Primordial and primary follicles were counted despite showing a nucleus.

## Analysis of liver

**Chemical analysis.**    The analysis of PCBs, OCPs, BFRs and PFASs was conducted on liver of dams (F1) and female offspring (F2) at the Laboratory of Environmental Toxicology, NMBU, Oslo, Norway. Samples of the left lobe were pooled according to their respective dose (Control, Low and High) and sampling time (pregnant, post-pregnant, and 3, 6 and 9 weeks of age). The method for PCB, OCP and BFR quantification is based on Brevik [45] and Polder and colleagues [46], and is accredited (except for BDE-206, -207, and -208) by the Norwegian Accreditation for chemical analysis in biota (requirements of the NS-EN ISO/IEC 17025, TEST 137). Quantification of PFASs is not accredited, but is validated according to the same procedures and quality control measures, and is described by Grønnestad and colleagues [47]. Details on method modification for the present study are elaborated in S1 File. Method quality control measures were approved as they were within accreditation requirements (see S1 File). Compounds detected at concentrations outside the accepted range of recovery (70–130%) were corrected for recovery (herein *p,p'*-dichlorodiphenyldichloroethylene (DDE), BDE-206 and HBCD). Concentrations below limit of detection (LOD) were replaced with the LOD to be included in the ΣPCBs, ΣOCPs, ΣPBDEs and ΣPFASs.

**Histopathological examination.**    Cross sections of the left lobe from dams (F1), and female and male offspring (F2) were embedded in paraffin after fixation. Sections were made (3 μm) using a microtome, transferred to microscope glass slides, and stained with HE. Examination was conducted by a board-certified pathologist under light microscope blinded by dose. Classification was based on the liver nomenclature guidelines recommended by the International Harmonization of Nomenclature and Diagnostic Criteria (INHAND) [48]. Preneoplastic and neoplastic lesions were noted as present (1) or absent (0). Non-neoplastic lesions were graded on a severity scale from 0 to 4 (0 = no, 1 = minimal, 2 = mild, 3 = moderate, and 4 = severe change) and included bile duct hyperplasia, fatty change (diffuse), extramedullary hematopoiesis, inflammation (chronic active), hepatocyte fatty change (diffuse), hepatocyte centrilobular hypertrophy, Ito cell hypertrophy, and oval cell hyperplasia.

**Liver microsomal preparation and cytochrome P450 activity analysis.**    Liver samples (right, caudate, and quadrate lobes) from female offspring (F2) were shipped (on dry ice) to the Department of Molecular Sciences, Swedish University of Agricultural Sciences, Uppsala, Sweden, for cytochrome P450 (CYP) activity analysis. Approximately 1 g tissue were used to prepare hepatic microsomes by a calcium aggregation method previously described [49]. Protein concentrations in the microsomes were measured with a commercially available kit (Bio-Rad Laboratories Inc., Hercules, USA) according to the manufacturer's instructions and using bovine serum albumin as standard. Microsomes were diluted to a protein concentration of 4 mg/mL and stored at -80˚C until use.

The activities of CYP1A and CYP2B10 were determined as a rate of 7-ethoxyresorufin (EROD, CYP1A1), 7-methoxyresorufin (MROD, CYP1A) and 7-pentoxyresorufin (PROD,

CYP2B10) O-dealkylation [50]. CYP3A11 activity were established as a rate of and benzyloxyre-sorufin (BROD) O-dealkylation [51]. CYP2E1 and CYP2A5 activities were determined as a rate of p-nitrophenol (PNPH) and coumarin-7- hydroxylation (CoH), respectively [52]. These iso-forms were selected because of their well-known role in xenobiotic metabolism. Experimental conditions for activity assays are reported in S1 File. Briefly, incubation mixtures were comprised of microsomal protein (0.2 mg for all enzymes except for PNPH with 0.5 mg), phosphate buffer (50 mM, pH 7.4) and the appropriate substrate (1 μM for EROD, 2 μM for MROD and BROD, 10 μM for PROD, and 200 μM for PNPH and CoH). Reactions were initiated by addition of 0.5 mM NADPH. The solutions were incubated in a water bath at 37°C for 5 (EROD), 7 (MROD and BROD), 15 (CoH), 20 (PROD), or 30 (PNPH) min. Reactions were terminated by adding 0.5 mL ice-cold methanol (40% TCA for PNPH) and the mixtures were centrifuged (7500 g at 4°C for 5 min). The amount of formed resorufin in the supernatants (for CYP1A, CYP2B10 and CYP3A11) was measured using HPLC with fluorescence detector (560 and 586 nm for excitation and emission wavelengths, respectively). The amount of formed p-nitrochate-col (for CYP2E1) was measured using HPLC with UV detector (345 nm). The amount of formed hydroxycoumarin (for CYP2A5) was measured using HPLC with fluorescence detector (338 and 458 nm for excitation and emission wavelengths, respectively). Formation of all metabolites were linear with microsomal protein concentrations and incubation times. All enzymatic activities were expressed as pmol of reaction product per mg protein and min.

## Statistical analysis

Box and bar plots were created using R Studio version 3.6.1 [53] and the packages 'ggplot2' [54] and 'ggpubr' [55]. Statistical analyses were conducted in JMP Pro 13® (SAS, Cary, USA) and a p-value $\leq 0.05$ was considered statistically significant.

The measurements of weights, the number of TEBs (3 and 6 weeks only) and the branching density were normally distributed and with satisfactory variance homogeneity. Thus, these variables were analyzed using multivariate linear regression by standard least square personality, generating least square mean values that were further analyzed using ANOVA. Body, liver and ovaries + uterus weights were analyzed within each sampling time and with dose and weight at weaning (dam body weight), number of offspring per litter (offspring body weight) or body weight (liver and ovaries + uterus weight) as explanatory variables. The number of TEBs (3 and 6 weeks only) and branching density were analyzed within each sampling time with dose and body weight as explanatory variables. P-values were generated by performing Dunnett's test on least square mean values.

Contingency table analysis using Chi-square likelihood ratio was conducted on the ratio between males and females, and the severity scores of hepatic histopathology within each sampling time and using the Steel-Dwass method to compare doses. The number of live offspring and ovarian follicles were analyzed for differences between doses within each sampling time by ANOVA followed by Dunnett's or Tukey HSD test, respectively. The MG development scoring, the number of TEBs (9 weeks), N, MEA, mammary epithelial length and glandular area, as well as the hepatic enzyme activities, were not normally distributed and analyzed for differences between doses within each sampling time by non-parametric comparison using the Steel or Steel-Dwass methods.

## Results

### Hepatic internal dosimetry

The concentrations of PCBs, OCPs, BFRs and PFASs were measured in one pooled sample of liver tissue from each dose in dams (F1) and female offspring (F2). Results are presented in

**Table 2. POP concentrations in pooled liver.**

| | Σ6PCBs[a] | Σ7OCPs[b] | *p,p'*-DDE | ΣBDE-28-183[c] | ΣBDE-206-209 | HBCD | ΣPFASs[d] |
|---|---|---|---|---|---|---|---|
| **Pregnant dams** | | | | | | | |
| Control | 87.1 | 9.4 | <LOD | 0.7 | 0.4 | <LOD | 2.8 |
| Low | 45691.6 | 2185.9 | 907.0 | 373.2 | 2260.7 | <LOD | 645.3 |
| High | 87012.6 | 32585.7 | 13883.9 | 6923.4 | 35303.5 | <LOD | 16183.3 |
| **Post-pregnant dams** | | | | | | | |
| Control | 21.4 | 4.2 | <LOD | 0.7 | 7.9 | <LOD | 6.5 |
| Low | 7787.9 | 802.6 | 922.2 | 345.5 | 4534.1 | <LOD | 1248.5 |
| High | 52743.1 | 15343.4 | 14189.0 | 8197.3 | 35411.8 | 40.2 | 22744.6 |
| **3 weeks offspring** | | | | | | | |
| Control | 48.8 | 12.3 | <LOD | 0.7 | 10.2 | <LOD | 32.2 |
| Low | 19629.9 | 1779.2 | 1200.9 | 515.5 | 883.1 | <LOD | 403.7 |
| High | 89968.8 | 38043.8 | 24412.2 | 11397.7 | 16274.1 | <LOD | 8535.0 |
| **6 weeks offspring** | | | | | | | |
| Control | 65.3 | 35.0 | <LOD | 0.7 | 0.4 | <LOD | 5.5 |
| Low | 10144.7 | 502.5 | 153.6 | 79.2 | 30.6 | <LOD | 170.9 |
| High | 50364.8 | 11392.5 | 1287.5 | 1102.7 | 293.9 | <LOD | 3490.1 |
| **9 weeks offspring** | | | | | | | |
| Control | 81.0 | 38.1 | <LOD | 0.7 | 4.1 | <LOD | 6.3 |
| Low | 5318.4 | 357.0 | 10.2 | 38.5 | 22.7 | 27.6 | 142.3 |
| High | 30278.5 | 5068.8 | 283.7 | 397.5 | 165.6 | <LOD | 2521.8 |

Concentrations of Σ6PCBs, Σ7OCPs, *p,p'*-DDE, PBDEs, HBCD and ΣPFASs in pooled liver samples from pregnant (gestation d 17) and post-pregnant (21 d post-partum) dams and female offspring (3, 6 and 9 weeks of age). Dams were dietary exposed to the mixture of POPs at Control, Low or High doses (0x, 5000x or 100 000x human estimated daily intake, respectively). Female offspring were exposed *in utero* and through lactation (ending at 3 wk). Values are presented as ng/g lipid weight for PCBs, OCPs, *p,p'*-DDE, PBDEs and HBCD, and ng/g wet weight for PFASs. Limit of detection (LOD), concentrations of individual chemicals and lipid (%) are presented in S1 File, and S1 and S2 Tables.

[a]Σ6PCBs: PCB-52, PCB-101, PCB118, PCB-138, PCB-153 and PCB-180.

[b]Σ7OCPs: HCB, α-chlordane, oxychlordane, *trans*-nonachlor, α-HCH, β-HCH and γ-HCH.

[c]ΣBDE-28-183: BDE-28, BDE-47, BDE-99, BDE-100, BDE-153, BDE-154 and BDE-183.

[d]ΣPFASs: PFHxS, PFOS, PFOA, PFNA, PFDA and PFUnDA.

Table 2 as Σ6PCBs, Σ7OCPs, *p,p'*-DDE, ΣBDE-28-183, ΣBDE-206-209, HBCD and ΣPFASs. Individual concentrations are presented in S1 (ng/g lipid) and S2 Tables (ng/g wet weight). LOD, recovery (%) and fat (%) are presented in S1 File and S2 Table.

Multiple POPs were detected in dams and offspring exposed to the Control dose. However, all compounds had lower concentrations compared to Low (6-5652x). Furthermore, both dams and offspring had higher concentrations of all compounds in High compared Low (2-25x). PCB-52 was not detected in any samples. BDE-183 and HBCD were only detected in one (post-pregnant dam, High dose) and two (post-pregnant dam, High dose, and 9 weeks offspring, Low dose) samples, respectively. Debromination of BDE-209 was seen in both dams and offspring.

Σ6PCBs and Σ7OCPs were slightly higher (1.6-6x) in pregnant dams compared to post-pregnant dams. On the other hand, ΣBDE-28-183, ΣBDE-206-209 and ΣPFASs were similar or 2x lower in pregnant dams compared to post-pregnant dams. All compounds (except PCB-52, BDE-183 and HBCD) were detected in 3-week-old offspring from the Low and High doses. Σ6PCBs, Σ7OCPs and ΣBDE-28-183 were 1.3–2.5x higher, while ΣBDE-206-209 and ΣPFASs were 2.2–5.1x lower, in 3-week-old offspring compared to post-pregnant dams. After end of

perinatal exposure, all compounds (except BDE-153 and HBCD) had decreasing concentrations with increasing age of offspring.

## Biometrical measurements

The body and liver weights of dams (F1) and offspring (F2) are presented in S3 Table. The POPs did not affect the body weight of dams (F1), female offspring (F2) or 30-week-old male offspring (F2). On the other hand, male offspring perinatally exposed to the Low dose had significantly higher body weights compared to controls at 9 weeks (p = 0.02).

The High dose caused a significant increase in liver weights of pregnant and post-pregnant dams (p < 0.01 and p = 0.05, respectively). In addition, the High dose increased the liver weight of 6-week-old female offspring (p = 0.02). No changes in liver weights were seen in 3, 9 and 30-week old offspring.

The number of live offspring and the ratio between male and female offspring were not significantly different from Control (S4 Table). However, the Low dose led to the production of more males while the High dose produced more females (p = 0.02). The collective uterus and ovary weights from 6 and 9-week-old female offspring were not significantly different from Control (S3 Table).

## Mammary gland histology

The effect of perinatal exposure to the mixture of POPs on MG development was investigated in female offspring (F2). Complete results are presented in S5 Table.

The development score was not affected by exposure. However, a trend towards less developed mammary glands was evident in the High dose at 3 weeks (p = 0.08). This trend was supported by a significantly decreased number of TEBs (High dose), compared to Control, at 3 weeks (p = 0.04). In addition, less TEBs were seen at 9 weeks in the Low dose, but this decrease was not significant from Control (p = 0.07). The number of TEBs was not affected by exposure at 6 weeks.

A visual difference in MG morphology between exposed and control mice was apparent, especially, for the High dose. At 6 weeks, the High dose had a significantly higher branching density compared to Control. An increased branching density was also apparent at 9 weeks, but not significantly different from Control (p = 0.1). The total number of epithelial intersections was not affected by perinatal POP exposure. On the other hand, the mammary epithelial area was reduced by the Low dose, compared to Control, at 6 and 9 weeks (p = 0.05 and p = 0.01, respectively). The epithelial length was also reduced by the Low dose at 9 weeks (p = 0.02). MG whole mounts representative of each dose and age are shown in Fig 1.

Histological examination of MGs did not reveal any significant effects of perinatal POP exposure. However, a trend towards more glandular tissue occupying the fat pad was seen by the Low dose at 6 weeks (p = 0.08).

## Ovarian histology

Alterations in ovarian follicles at different stages in folliculogenesis were investigated in female offspring (F2) after perinatal exposure to the mixture of POPs. Results are presented in Fig 2.

The number of primordial follicles was not affected by exposure. The Low dose significantly reduced the number of primary follicles at 3 weeks, compared to Control (p < 0.01), but not at 6 and 9 weeks. The number of preantral follicles did not differ in exposed mice compared to controls. However, the Low dose caused the formation of significantly less preantral follicles than the High dose at 6 (p = 0.01) and 9 weeks (p < 0.01), with the Control showing an

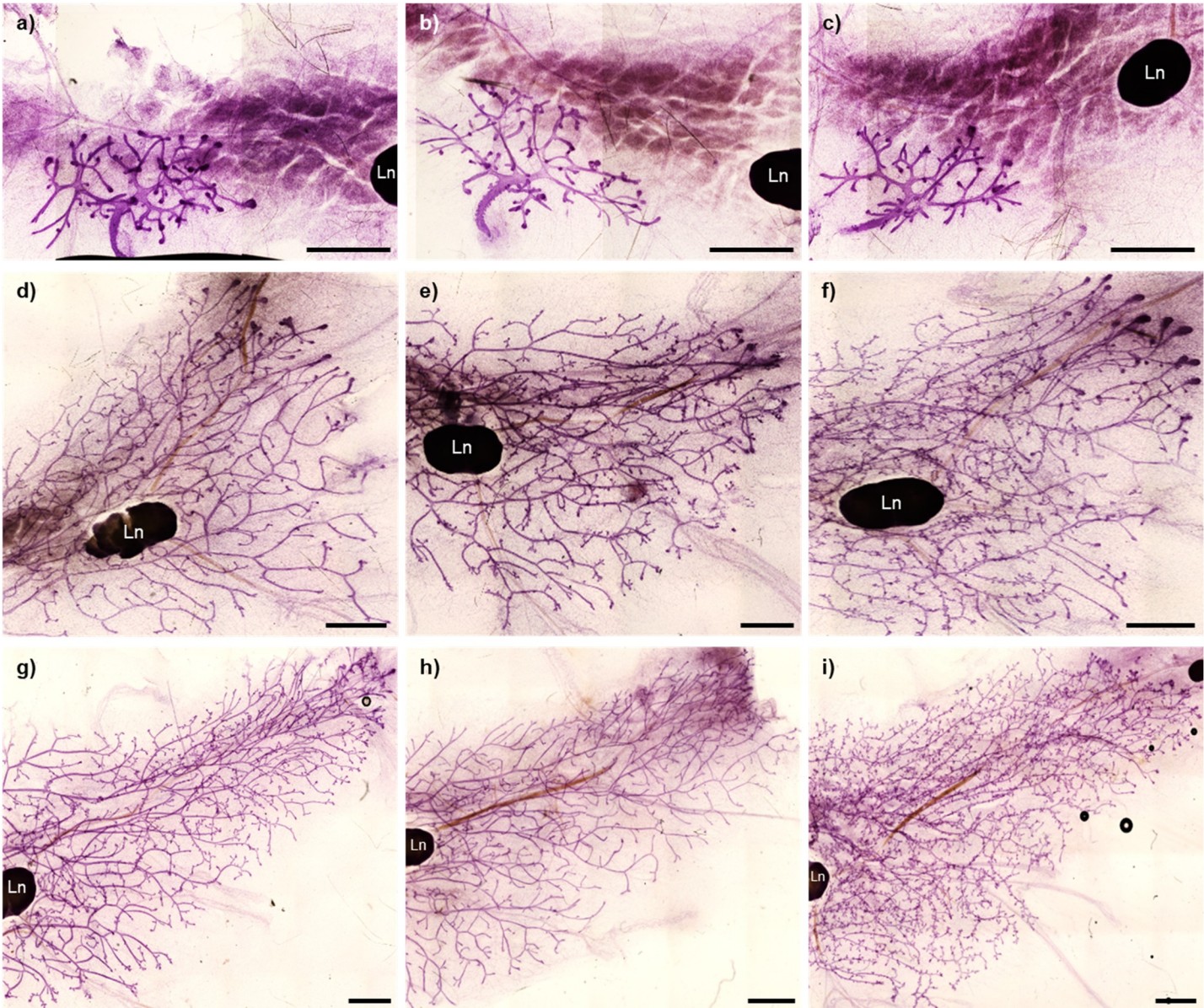

**Fig 1. Mammary gland whole mounts.** Whole mounts from 3 (a, b and c), 6 (d, e and f) and 9-week-old (g, h and i) female offspring maternally exposed to a mixture of POPs at Control, Low or High doses (0x, 5000x or 100 000x human estimated daily intake, respectively). Pictures illustrate the morphology representative of each dose; Control (a, d, and g), Low (b, e, and h) or High (c, f, and i). Each whole mount is composed of multiple images taken at 10-fold magnification and put together as a mosaic using ImageJ software. Scale bar = 2000 μm. Ln = Lymph node.

intermediate follicle number. The High dose reduced the number of antral follicles at 3 weeks ($p < 0.01$), but not at 6 weeks, compared to Control. At 9 weeks, the number of antral follicles were lower than but not significantly different from Control (with $p = 0.06$ and $p = 0.07$ for High and Low, respectively).

## Hepatic histology

Histopathological examination was conducted on livers from dams (F1) and offspring (F2) dietary or perinatally exposed to the mixture of POPs. No pre-neoplastic or neoplastic lesions

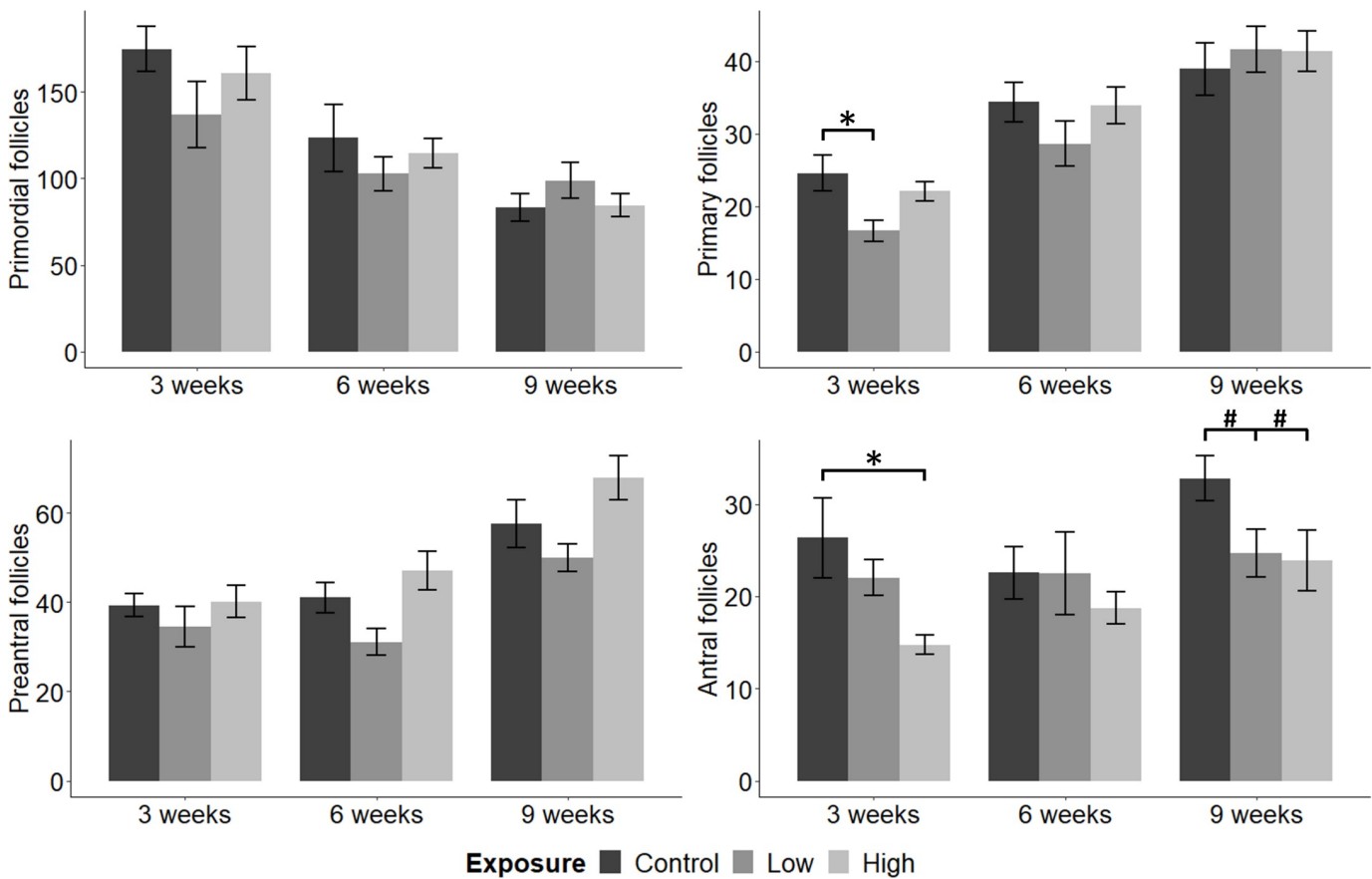

**Fig 2. Ovarian follicle numbers.** The number of primordial, primary, preantral and antral follicles in ovaries of 3, 6 and 9-week-old female offspring maternally exposed to a mixture of POPs at Control, Low or High doses (0x, 5000x or 100 000x human estimated daily intake, respectively). Histopathological examination was conducted on every 10th ovarian section. Results are presented as mean ± standard error. 3 weeks: n = 9, 11 and 12 for Control, Low and High, respectively. 6 weeks: n = 11, 10, and 11 for Control, Low and High, respectively. 9 weeks: n = 10, 11 and 9 for Control, Low and High, respectively. *p-values ≤ 0.05. #p-values ≤ 0.1.

were noted. Focal fatty lesions were seen in 5 9-week-old female offspring perinatally exposed to the High dose but did not differ significantly from Control.

Dietary exposure to the Low and High doses increased the severity of hepatocellular hypertrophy (of the centrilobular region) in pregnant and post-pregnant dams compared to Control (p < 0.01). Perinatal exposure to the High dose caused hypertrophy (with increased severity) in female offspring at 3 weeks (p = 0.02). At 6 and 9 weeks, both the Low and High doses caused hepatocellular hypertrophy with increased severity from Control (p < 0.01). Hepatocellular hypertrophy was also seen in 9-week-old male offspring with a higher severity than Control in both the Low and High doses (p < 0.01). This hypertrophy persisted until 30 weeks in male offspring, which had a significantly higher severity compared to Control (Low dose only, p < 0.01). Fig 3 illustrates the hepatocellular hypertrophy representative of each dose in female offspring at 9 weeks.

Extramedullary hematopoiesis was seen in female offspring at 3, 6 and 9 weeks with decreasing incidence and severity with increasing age and was not affected by perinatal exposure. Mean severity scores of the hepatocellular hypertrophy and hematopoiesis (± standard error) in dams and offspring are presented in S6 Table. No other non-neoplastic lesions were seen.

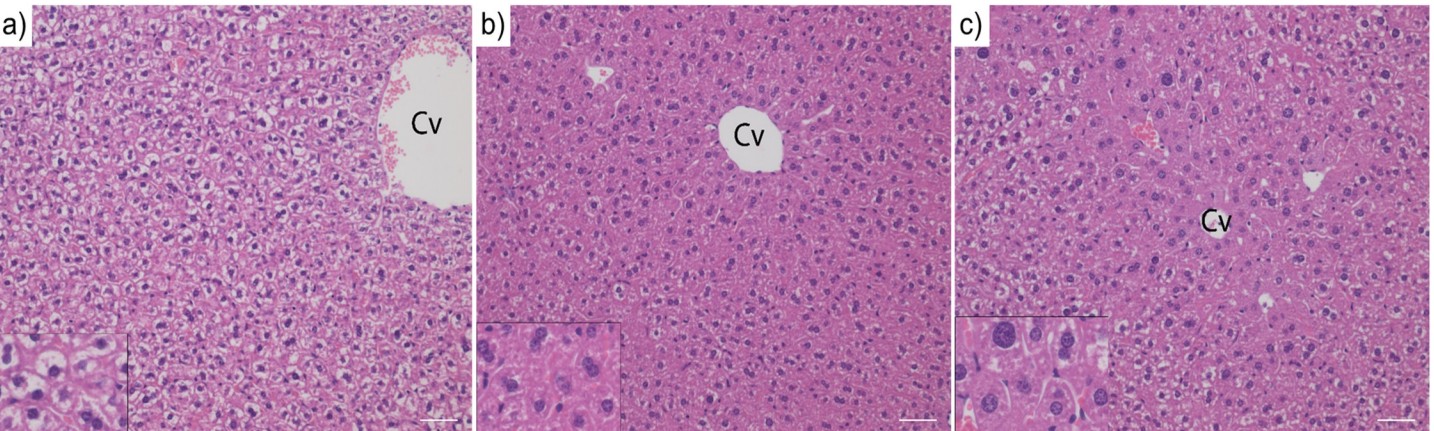

**Fig 3. Hepatic morphology.** Representative hepatic morphology of 9-week-old female offspring maternally exposed to a mixture of POPs at Control (a), Low (b) or High (c) doses (0x, 5000x or 100 000x human estimated daily intake, respectively). Hepatocytes have increased size and frequently enlarged hyperchromatic nuclei, which were graded on a scale from 0 to 4 (0 = no, 1 = minimal, 2 = mild, 3 = moderate, and 4 = severe change). The Low (b) and High (c) doses exhibited mild and severe hypertrophy, respectively, compared to the normal morphology of the Control (a). Magnification at 200-fold (inset at 400-fold). Scale bar = 50 μm. Cv = Central vein.

### Hepatic cytochrome P450 activity

The activities of hepatic CYP enzymes were measured in female offspring (F2) perinatally exposed to the mixture of POPs (Fig 4).

Perinatal exposure significantly induced the activity of all CYPs (except CYP2A5) at 3 weeks ($p < 0.01$ for all, except for CYP1A for the Low dose with $p = 0.05$), with the High dose inducing the largest change from Control for CYP1A1, CYP1A, CYP3A11 and CYP2B10. For CYP2E1, the activity was decreased by the Low dose, and increased by the High dose, compared to Control ($p < 0.01$).

At 6 weeks, the activities of all CYPs were affected by the High dose ($p \leq 0.01$, except for CYP2A5 with $p = 0.02$). Furthermore, the Low dose caused higher activities of CYP1A1 ($p = 0.02$), CYP3A11 ($p < 0.01$) and CYP2B10 ($p < 0.01$) compared to Control. The activity of CYP2E1 was reduced by both the Low and High doses ($p < 0.01$). The Low dose also reduced the activity of CYP2A5 ($p < 0.01$), while the High dose increased the CYP2A5 activity compared to Control ($p = 0.02$).

At 9 weeks, the CYP1A1 and CYP1A activities were no longer affected by perinatal exposure. However, the High dose still increased the activity of CYP3A11 and CYP2B10 compared to Control ($p < 0.01$). In addition, the Low dose elevated the activity of CYP2B10 ($p < 0.01$). On the other hand, the Low dose reduced the activities of CYP2E1 and CYP2A5 compared to Control ($p < 0.01$).

### Discussion

The present study demonstrated that *in utero* and lactational exposure to the mixture of POPs altered MG development and ovarian follicle maturation and caused persistent hepatocellular hypertrophy and enzyme induction in CD-1 mice. In dams, the POPs (apart from PCB-52 and HBCD) were readily taken up from the diet and distributed to the hepatic tissue. PCBs and OCPs seemed to be more efficiently transferred to offspring during late gestation and lactation compared to BFRs and PFASs. However, all compounds (except PCB-52, BDE-183 and HBCD) were detected in weaning offspring, thus, confirming maternal transfer of the mixture. After the end of lactational exposure, the POPs decreased in concentration over time suggesting dilution due to weight gain, metabolization and excretion.

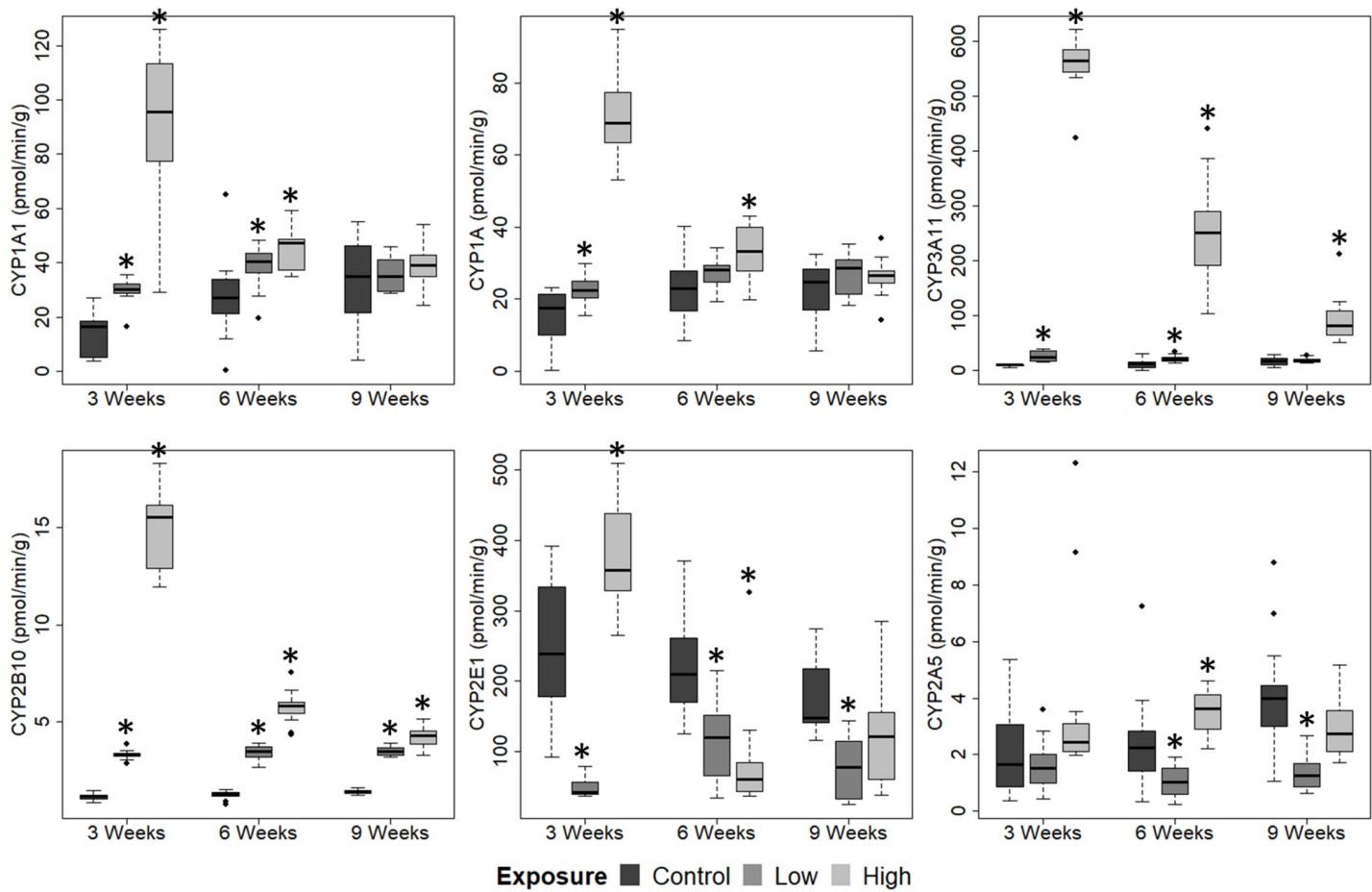

**Fig 4. Hepatic enzymatic activity.** The activities of cytochrome P450 (CYP) 1A1, 1A, 3A11, 2B10, 2E1 and 2A5 in female offspring maternally exposed to a mixture of POPs at Control, Low or High doses (0x, 5000x or 100 000x human estimated daily intake, respectively). Samples were taken at 3 (n = 12), 6 (n = 14) and 9 (n = 14) weeks of age. Activities were measured as a rate of 7-ethoxyresorufin (CYP1A1), 7-methoxyresorufin (CYP1A), benzyloxyresorufin (CYP3A11), 7-pentoxyresorufin (CYP2B10) dealkylation, and p-nitrophenol (CYP2E1) and coumarin-7 (CYP2A5) hydroxylation. Boxplots consist of 25th, 50th (median) and 75th percentiles, whiskers (extending to 1.5 interquartile range) and outliers (•). Significant difference from Control (p ≤ 0.05) is marked by *.

Previously, Johanson and colleagues [8] showed that perinatal exposure to the Low dose resulted in 2-35x higher POP concentrations in 20-week-old A/J Min/+ mice (liver and adipose tissue) than the average blood levels reported in the Scandinavian population [39]. In the present study, Low dose female offspring at 9 weeks had comparable (from 2x lower to 5x higher) lipid-adjusted concentrations to those found previously [8]. This may suggest slight strain-dependent difference in metabolism and excretion of POPs and that the Low dose could be considered relevant to humans.

Results from the traditional qualitative scoring of MG development in female offspring showed a trend towards reduced development in 3-weeks-old mice perinatally exposed to the High dose. In addition, the number of TEBs were significantly lower in these mice. As previously described [12], MG development can be regarded as accelerated if the number of TEBs in the treated group is higher than that of the control at 3 weeks of age. Exposure to hexachlorobenzene (HCB) has been shown to increase the number of TEBs in C57BL/6 mice [56], which could increase the risk of breast cancer [57–59]. However, the opposite was observed in the present study and, thus, perinatal exposure to the High dose of POPs seemed to restrict MG development in 3-week-old female mice.

The number of TEBs is also coincident with the number of epithelial ducts in the MG. A gland with a high branching density will, thus, have a higher amount of TEBs [12]. Although the number of TEBs was not affected by perinatal exposure at 6 and 9 weeks, the High dose increased the branching density significantly in pubertal and non-significantly in adult mice. This could suggest a retention of TEBs causing more branching and possibly indicate a continuation of the restricted MG development observed in 3-week-old mice. Furthermore, perinatal exposure to the Low dose decreased the size and length of the epithelial expansion throughout the MG in pubertal and adult mice, which suggested that the more human relevant dose of POPs may have prematurely arrested MG development.

Restricted MG development has previously been shown following *in utero* exposure to PFOA at concentrations as low as 0.01 mg/kg in CD-1 and C57BL/6 mice [15, 16, 18, 19]. Interestingly, the delayed gland development found by Macon and colleagues [18] was evident at lower doses of PFOA than those required to increase the liver weight and persisted for a relatively long time after end of exposure. *In utero* and lactational exposure to DE-71 (a mixture of PBDEs) also delayed MG development in 3-week-old female Long-Evans rats [60]. Furthermore, *in utero* exposures to other EDCs, such as 2,3,7,8-Tetrachlorodibenzo-*p*-dioxin (TCDD), Bisphenol A and Atrazine metabolites, have been shown to restrict gland development in rats and mice [61–63]. However, as previously emphasized [17], caution should be taken when drawing conclusions from studies using a single mouse strain as differences have been observed in the sensitivity of strains to PFOA-induced alterations in MG development [17, 19].

Overall, perinatal exposure to the mixture of POPs affected ovarian folliculogenesis in CD-1 mice. The Low dose reduced the number of primary follicles, while the High dose reduced the number of antral follicles in weaning offspring. In addition, there was a non-significant decrease in the number of antral follicles in adult offspring. This may suggest a delayed rate of follicle maturation. Several previous studies exposing rats to single POPs (including TCDD, PCB-126, BDE-99, BDE-47, PFOA and perfluorooctanesulfonic acid (PFOS)) during early development have also reported delays in ovarian follicle maturation [64–69]. Furthermore, evidence indicates that POPs can disrupt ovarian folliculogenesis by directly affecting the ovary, by modulating endocrine pathways or by altering circulating hormone levels [66, 68, 70]. After primordial follicle assembly, the initial recruitment is triggered by stimulation from intraovarian and/or unknown factors promoting follicle growth into primary, preantral and antral follicles. Most antral follicles undergo atresia. However, when the optimal stimulus is present, a few follicles continue into cyclic recruitment and reach the preovulatory stage. This process is controlled and stimulated by gonadotropins (e.g. luteinizing hormone and follicle-stimulating hormone) that have been synthesized and excreted by the anterior pituitary. Gonadotropin production is regulated by the gonadotropin releasing hormone (released from the hypothalamus), which further is controlled by ovarian steroid hormones that have been synthesized in response to gonadotropin stimulation. This is known as a negative feedback loop which tightly controls follicle maturation [71, 72]. Due to the endocrine disrupting properties of many POPs, it is likely that perinatal exposure may have disturbed the hormonal regulation of ovarian folliculogenesis [73]. However, alterations in hormone levels were not investigated in the present study.

The reduced number of antral follicles could also suggest an increased rate of follicle atresia. However, the number of atretic follicles were not quantified in the present study. An increased number of atretic follicles has previously been reported following perinatal exposure to a mixture of PCB-101 and -118 in CD-1 mice [74] and *in utero* exposure to PCB-126 in Sprague-Dawley rats [65]. Increased follicle atresia can result in an acceleration of ovarian reserve depletion and induce premature ovarian failure, thus, shortening the reproductive lifespan

[20, 75]. In humans, multiple OCPs, PCBs and BFRs have been associated with premature menopause [76–80]. Unfortunately, no measures were taken to promote estrous cycle homogeneity in mice. Consequently, the pubertal and adult offspring may have been terminated at different stages of estrous cycle and possibly affecting the ovarian follicle numbers.

Dietary exposure to the High dose increased the relative liver weight in dams, an increase possibly caused by hepatocellular hypertrophy. In addition, hypertrophy was seen in dams given the Low dose proving the sensitivity of this endpoint in CD-1 mice exposed to POPs. Multiple other studies have also showed hepatocellular hypertrophy in response to POP exposure [17, 34, 81–86]. Hypertrophy of liver cells is often seen together with hyperplasia and increased enzymatic activity. Hyperplasia was not detected in the present study. However, dose-dependent increases in hepatic CYP activities were seen in female offspring following perinatal exposure to the mixture of POPs. Interestingly, combined hepatic hypertrophy, hyperplasia and heightened enzymatic activities have previously been evaluated as predictors of liver cancer [26, 87, 88].

Increased CYP activities may lead to the formation of reactive oxygen species and cause hepatotoxicity [28, 29, 89]. In addition, the enhanced activities of CYP1A, CYP3A and CYP2B are commonly used as biomarkers for AhR, PXR and CAR activation, respectively [31, 90–93]. In the present study, CYP1A1 and CYP1A, CYP3A11 and CYP2B10 were dose-dependently induced in offspring, indirectly indicating AhR, PXR and CAR activation. Furthermore, the induction was less prominent with time corresponding to the temporal decline in hepatic POP concentrations. Interestingly, the activities of CYP2E1 and CYP2A5 were (at some points) lower in perinatally exposed mice compared to controls. This may have been caused by a reduced capacity for xenobiotic metabolism after POP exposure during early development. Physiological consequences of this should be further investigated, as low activity of these enzymes might predispose animals to harmful health effects under conditions where CYP2E1 and CYP2A5 are required for detoxification.

Previously, alterations in rodent hepatic CYP activities have been shown following perinatal exposure to POPs, such as PBDEs [33, 36, 94, 95], γ-hexachlorocyclohexane (HCH) [96] and mixtures [97–99]. Other studies have also observed hepatocellular hypertrophy caused by *in utero* and lactational exposure to mixtures of POPs [36, 84, 98, 99] or single chemicals [36, 100, 101]. Generally, resolution of hepatocellular effects occurs when the chemical has been metabolized and excreted from the system [26]. This is dependent on the nature of the chemical and highly persistent compounds (e.g. POPs) may, thus, cause effects long after end of exposure. The persistent nature of the Low dose has recently been demonstrated with higher (< 136x) levels of POPs in A/J Min/+ mice 17 weeks after end of perinatal exposure to the mixture of POPs [8]. Furthermore, the present study showed sustained hepatocellular hypertrophy with a significantly increased severity in 30-week-old male offspring perinatally exposed to the Low dose compared to controls.

Notably, perinatal exposure to the Low dose increased the body weight of male but not female offspring at 9 weeks. Previously, perinatal exposure to BDE-47 elevated the body weight of male but not female Sprague-Dawley rats [102] and stimulated the development of obesity in male ICR mice [103]. Various POPs have also been associated with increased weight gain and obesity in humans [104–107]. Thus, the increased body weight of male mice observed could possibly suggest an obesogenic effect of developmental exposures to POPs. No other effects were seen on body weight in both dams and offspring. This could be explained by POP concentrations generally being below the no observed adverse effect level (NOAEL) for individual chemicals [39], as well as the differences in xenobiotic metabolism between mice and humans [108, 109].

In summary, the present study illustrated that perinatal exposure to a human relevant mixture of POPs modulated female mammary gland development and ovarian folliculogenesis in

CD-1 mice. Furthermore, the mixture caused persistent hepatocellular hypertrophy, enzymatic induction and slight liver enlargement suggesting modest hepatotoxic effects. Further studies should include measurements of multiple hormones to create a better understanding of the effects observed on the development of the female reproductive organs. Additionally, studies should focus on how human relevant mixtures of POPs can ultimately affect female reproduction or negatively impact liver function.

## Supporting information

**S1 Fig. Study design of the experiment.** Timed-pregnant CD-1 dams (F0) gave birth to F1 dams at approximately 1 week after arrival. F1 dams were exposed to the mixture of POPs through the diet at Control, Low or High doses (0x, 5000x or 100 000x human estimated daily intake, respectively) from 3 weeks of age until termination during pregnancy (gestation d 17) or post-pregnancy (d 21 post-partum). F2 offspring were only exposed to the mixture *in utero*, through lactation and by nibbling on their mother's food prior to weaning, and were terminated at 3 (females), 6 (females), 9 (females and males) or 30 (males) weeks of age.
(TIF)

**S1 File. Supporting information on materials and methods.**
(DOCX)

**S1 Table. Lipid adjusted concentrations of POPs.** Individual PCBs, OCPs, BFRs and PFASs in pooled liver samples of dietary exposed pregnant (gestation d 17) and post-pregnant (21 d post-partum) dams and maternally exposed female offspring (3, 6 and 9 weeks of age). Mice were exposed to the mixture of POPs at Control, Low or High doses (0x, 5000x or 100 000x human estimated daily intake, respectively). Values are presented as ng/g lipid weight for PCBs, OCPs and BFRs, and ng/g wet weight for PFASs.
(DOCX)

**S2 Table. Lipid percentage and wet weight concentrations of POPs.** PCBs, OCPs, BFRs and PFASs (ng/g wet weight) and lipid (%) in pooled liver samples of dietary exposed pregnant (gestation d 17) and post-pregnant (21 d post-partum) dams and maternally exposed female offspring (3, 6 and 9 weeks of age). Mice were exposed to the mixture of POPs at Control, Low or High doses (0x, 5000x or 100 000x human estimated daily intake, respectively).
(DOCX)

**S3 Table. Biometrical measurements.** Body and liver weight of pregnant and post-pregnant dams, and female (3, 6 and 9 weeks of age) and male (9 and 30 weeks of age) offspring. Combined ovaries and uterus weight of female offspring (6 and 9 weeks of age) is also presented. Dams were dietary exposed to a mixture of POPs at Control, Low or High doses (0x, 5000x or 100 000x human estimated daily intake, respectively). Offspring were exposed in utero and through lactation (ending at 3 weeks). Pregnant dams were euthanized on gestation d 17 (n = 12, 16 and 8 for Control, Low and High, respectively). Post-pregnant dams were euthanized 21 d post-partum (n = 14, 10 and n = 11 for Control, Low and High, respectively). Females 3 weeks: n = 12 (all groups). Females 6 and 9 weeks: n = 14 (all groups, except for ovary + uterus weight from 6 weeks Control where n = 13). Males 9 and 30 weeks: n = 15 (all groups, except for liver weight from 9 weeks Low where n = 14). Results are presented as least square mean ± standard error. Bold marks significant differences (p ≤ 0.05) from Control.
(DOCX)

**S4 Table. Number of live offspring, males, and females.** The number of live offspring produced by dams during dietary exposure to a mixture of POPs at Control, Low or High doses

(0x, 5000x or 100 000x human estimated daily intake, respectively) were counted on gestation d 17. Live offspring of the post-pregnant dams were counted, and gender was determined, at 21 d post-partum. Results are presented as mean ± standard error. The numbers of males/ females are given as the total number within each dose.
(DOCX)

**S5 Table. Complete results on mammary gland histology.** Mammary gland morphology in 3, 6 and 9-week-old female CD-1 mice maternally exposed to a mixture of POPs at Control, Low or High doses (0x, 5000x or 100 000x human estimated daily intake, respectively). Qualitative development scores were conducted on whole mounts of the 4th mammary gland from the right side using a scale from 1 to 4 (1 = poor development, 4 = best development). Terminal end buds (TEBs) were defined as ends with a diameter of $\geq$ 100 μm. Branching density, sum of the number of intersections, mammary epithelial area and length were calculated using Sholl analysis in ImageJ software. Glandular area was calculated using ImageJ on one section of fixated 4th mammary gland from the left side. Results are presented as mean ± standard error. At 3 weeks of age, n = 11 for the High group and n = 12 for the Control and Low groups except glandular area were n = 11. At 6 weeks, n = 12 for the Control group and n = 14 for the Low and High groups except glandular area were n = 12. At 9 weeks, n = 13 for the Control and High groups, except Control group glandular area were n = 12. Furthermore, n = 14 for the Low group except glandular area were n = 12. Numbers in bold mark significant difference from Control ($p \leq 0.05$). P-values $\leq 0.10$ are marked with *.
(DOCX)

**S6 Table. Hepatic extramedullary hematopoiesis and centrilobular hypertrophy.** Severity of extramedullary hematopoiesis and centrilobular hypertrophy of hepatocytes in livers from dams (pregnant sampled at gestation d 17, and post-pregnant sampled at 21 d post-partum), and female (sampled at 3, 6 and 9 weeks of age) and male (sampled at 9 and 30 weeks of age) offspring. Dams were dietary exposed, and offspring were maternally exposed, to a mixture of POPs at Control, Low or High doses (0x, 5000x or 100 000x human estimated daily intake, respectively). Results are presented as mean ± standard error. Severity was graded on a scale from 0 to 4 (0 = no, 1 = minimal, 2 = mild, 3 = moderate, and 4 = severe change). n = 12, 14 and 14 for female offspring in all groups at 3, 6 and 9 weeks, respectively. n = 15 for male offspring in all groups at both sampling times. n = 12, 16 and 8, and n = 14, 10 and 11 for the Control, Low and High groups of pregnant and post-pregnant dams, respectively. Bold indicates significant ($p \leq 0.05$) difference from Control.
(DOCX)

## Acknowledgments

The authors would like to thank the staff at the Section for Experimental Biomedicine for excellent technical assistance during the experiment.

## Author Contributions

**Conceptualization:** Erik Ropstad.

**Data curation:** Silje Modahl Johanson.

**Formal analysis:** Silje Modahl Johanson, Erik Ropstad, Gunn Charlotte Østby, Mona Aleksandersen, Galia Zamaratskaia, Gudrun Seeberg Boge, Ruth Halsne, Jan Ludvig Lyche, Steven Verhaegen.

**Funding acquisition:** Erik Ropstad.

**Investigation:** Gunn Charlotte Østby, Gudrun Seeberg Boge, Ruth Halsne, Cathrine Trangerud.

**Methodology:** Silje Modahl Johanson, Erik Ropstad.

**Project administration:** Erik Ropstad.

**Resources:** Gunn Charlotte Østby, Mona Aleksandersen, Galia Zamaratskaia, Hanne Friis Berntsen, Karin Elisabeth Zimmer.

**Supervision:** Erik Ropstad, Steven Verhaegen.

**Visualization:** Silje Modahl Johanson, Steven Verhaegen.

**Writing – original draft:** Silje Modahl Johanson.

**Writing – review & editing:** Erik Ropstad, Galia Zamaratskaia, Jan Ludvig Lyche, Hanne Friis Berntsen, Steven Verhaegen.

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
