## [Decision Letter · Decision Letter 0]

3 Feb 2021

PONE-D-20-37867

Perinatal exposure to a human relevant mixture of persistent organic pollutants: effects on mammary gland development, ovarian folliculogenesis and liver in CD-1 mice

PLOS ONE

Dear Dr. Johanson,

Thank you for submitting your manuscript to PLOS ONE. After careful consideration, we feel that it has merit but does not fully meet PLOS ONE’s publication criteria as it currently stands. Therefore, we invite you to submit a revised version of the manuscript that addresses the points raised by the both external experts during the review process.

Please note that one the reviewer is really reluctant about the novelty of this paper and quite negative for the fact that no endocrine data is available (mainly hormone levels). This lack of hormone measurement, even discussed in the paper, is clearly an important concern as the work regards endocrine disrupting chemicals and their effects on the development and functioning of hormone-dependent organs (mammary gland and ovary).

Altogether, I urge you to deeply respond to each comments. Be aware that complementary experiments may probably be necessary to fully answer these comments, specially for Reviewer 2.

We look forward to receiving your revised manuscript.

Kind regards,

Jean-Marc A Lobaccaro, PhD

Academic Editor

PLOS ONE

Journal Requirements:

3. We noticed you have some minor occurrence of overlapping text with the following previous publication, which needs to be addressed:

- https://onlinelibrary.wiley.com/doi/abs/10.1002/tox.20679

The text that needs to be addressed involves the sixth paragraph of the Discussion.

In your revision ensure you cite all your sources (including your own works), and quote or rephrase any duplicated text outside the methods section. Further consideration is dependent on these concerns being addressed.

Reviewers' comments:

Reviewer's Responses to Questions

**Comments to the Author**

1. Is the manuscript technically sound, and do the data support the conclusions?

Reviewer #1: Yes

Reviewer #2: No

2. Has the statistical analysis been performed appropriately and rigorously? 

Reviewer #1: Yes

Reviewer #2: I Don't Know

3. Have the authors made all data underlying the findings in their manuscript fully available?

Reviewer #1: Yes

Reviewer #2: No

4. Is the manuscript presented in an intelligible fashion and written in standard English?

Reviewer #1: Yes

Reviewer #2: No

5. Review Comments to the Author

Reviewer #1: In this paper, the authors have investigated the effects of a perinatal exposure to a mixture of POPs on mammary gland and ovarian folliculogenesis in mice.

The design of the study is good, in accordance with OECD guidelines.

The authors chose only 2 doses: low vs high. It could be considered as restrictive but the design of a F2 study is clearly too complex to consider more doses to investigate.

They well discussed the limits and the bias of their work, especially regarding the ovarian function (no estrous cycle homogeneity, no quantification of follicle atresia, no evaluation of hormone levels).

Some interrogations remain to consider to my opinion, and could enhance the discussion of the paper :

- how the authors can explain the lack of effect of high dose on BW of animals ?

- how they can explain the no detection of HBCD in dams liver ?

To my opinion, the authors must moderate their conclusion on the role on carcinogenesis, as their results did not well illustrate this point

In the introduction of the paper, the authors cited a lot of papers (55 !). They should reduce this number and add some publications regarding humans.

Reviewer #2: This study evaluates the impact of POP mixtures on mammary gland development, ovarian physiology and liver toxicity. However, it lacks novelty. Lack of hormonal analyses to support the results.

The conclusions are based on speculations or hypotheses.

6. PLOS authors have the option to publish the peer review history of their article (what does this mean?). If published, this will include your full peer review and any attached files.

Reviewer #1: No

Reviewer #2: No

---

## [Author Response · Author response to Decision Letter 0]

4 May 2021

Dear Academic Editor and Reviewers,

The authors thank you for taking the time to go through the manuscript and for providing comments that have improved the quality of the manuscript. Each comment and question have been addressed below.

Sincerely,

Dr. Silje Modahl Johanson

Reviewers' comments:

1. Is the manuscript technically sound, and do the data support the conclusions?

Reviewer #1: Yes

Reviewer #2: No

The authors thank the reviewers for taking the time to go through the manuscript. We agree that the conclusions should be moderated, particularly on the role on carcinogenesis. The manuscript has been edited accordingly. 

2. Has the statistical analysis been performed appropriately and rigorously? 

Reviewer #1: Yes

Reviewer #2: I Don't Know

The statistical analyses have been performed to the best of our abilities and we hope the reviewers are satisfied with the description presented in the manuscript.

3. Have the authors made all data underlying the findings in their manuscript fully available?

Reviewer #1: Yes

Reviewer #2: No

For additional transparency, all raw data have been deposited in the online repository DataverseNO and can be found at https://dataverse.no/privateurl.xhtml?token=e2d40b17-f2dc-4974-8e6e-d0a44d5ffbe7. This is a private URL that can be used during the reviewing process and the dataset will be given the DOI https://doi.org/10.18710/MSTY0R when and if the manuscript is accepted for publishing. The deposited dataset is identical to ‘S9 Raw data’ originally submitted for review.

4. Is the manuscript presented in an intelligible fashion and written in standard English?

Reviewer #1: Yes

Reviewer #2: No

The authors have had the manuscript checked by a native English speaker and we hope the reviewers now find it presented in an intelligible fashion and written in standard English. 

5. Review Comments to the Author

Reviewer #1: 

In this paper, the authors have investigated the effects of a perinatal exposure to a mixture of POPs on mammary gland and ovarian folliculogenesis in mice. The design of the study is good, in accordance with OECD guidelines. The authors chose only 2 doses: low vs high. It could be considered as restrictive but the design of a F2 study is clearly too complex to consider more doses to investigate. They well discussed the limits and the bias of their work, especially regarding the ovarian function (no estrous cycle homogeneity, no quantification of follicle atresia, no evaluation of hormone levels).

The authors thank the reviewer for taking the time to go through the manuscript and for the positive feedback. 

Some interrogations remain to consider to my opinion, and could enhance the discussion of the paper:

- how the authors can explain the lack of effect of high dose on BW of animals?

It is not clear why we did not observe any effect of the high dose on body weight. However, as the CD-1 mouse strain is an outbred laboratory strain it generally has a higher baseline variation in response to stimuli than inbred mouse strains. This could have caused the possible negative effects on body weight to be masked by natural variation and, thus, not showed significant differences here. Furthermore, the individual concentrations of POPs in the High dose were predominantly lower than the no observed adverse effect levels (NOAEL), where such levels were available [1]. Thus, despite this being a High dose compared to human consumption, the relatively more rapid xenobiotic metabolism of mice compared to humans could possibly explain the lack of toxic effect observed. A short discussion has been added to the manuscript on page 30, lines 611-615 in the Revised Manuscript with Track changes. 

- how they can explain the no detection of HBCD in dams liver?

The LOD of HBCD was relatively high (1.182 ng/g wet weight) compared to the detection limit of the other POPs. Thus, it is possible that some of the samples had concentrations of HBCD just below the LOD. Furthermore, as the chemical analysis was conducted only on one pooled liver sample from each generation, gender and exposure group, it is possible that some of the individual samples had higher HBCD concentrations than others, and, when pooling the samples, the overall concentration was diluted by samples with lower HBCD concentrations. 

The commercial mixture of HBCD consists of the three main diastereomers α, β and γ-HBCD, with the γ-HBCD being the predominating diastereomer comprising ≥ 70% [2]. In adult female mice, γ-HBCD has been shown to be rapidly metabolized and eliminated with a terminal half-life of approximately 4 days [3]. Thus, particularly for the offspring, it is possible that most HBCD had been eliminated from the body prior to sampling. However, this does not explain why dam livers continuously exposed to the mixture of POPs did not show higher levels of HBCD. Thus, the reason for the lack of HBCD in dam livers is unknown. 

To my opinion, the authors must moderate their conclusion on the role on carcinogenesis, as their results did not well illustrate this point. 

The authors agree with the reviewer and have moderated the conclusions on the role of the mixture of POPs on carcinogenesis. This can be seen on page 5 (line 55), page 28-29 (lines 571-575) and page 31 (line 624) in the Revised Manuscript with Track Changes.

In the introduction of the paper, the authors cited a lot of papers (55!). They should reduce this number and add some publications regarding humans.

The number of cited papers in the introduction has been reduced to 39 and more emphasis has been put on the Scientific Statements of The Endocrine Society [4, 5], which extensively reviews the existing literature on how exposure to EDCs may lay the foundation for various diseases later in life in both animals and humans. 

Reviewer #2: 

This study evaluates the impact of POP mixtures on mammary gland development, ovarian physiology and liver toxicity. However, it lacks novelty. Lack of hormonal analyses to support the results. The conclusions are based on speculations or hypotheses.

The authors thank the reviewer for taking the time to go through the manuscript. 

In the study we utilized a mixture of POPs that is highly relevant to humans due to its complexity and inclusion of compounds from multiple chemical classes. Furthermore, we applied a human relevant route of exposure and discovered effects on multiple organs that could lead to adverse effects later in life. Other research articles using the same mixture of POPs have been well received in the scientific community [6-9]. To the authors knowledge, no other study has previously presented simultaneous results on female reproductive organs and liver caused by exposure to a large mixture of POPs.

The authors strongly agree that hormonal analyses should have been conducted to support the results. However, due to the design of the experiment, only single hormone measurements would have been possible. This would not have given a thorough characterization of the impact of the mixture of POPs on hormone levels, or the impact of hormones on mammary gland development. A better option would have been to chemically synchronize estrous cycle of all female offspring. However, this could possibly have compromised the aim of the experiment and concealed the effects of the mixture of POPs. Another option would have been to use vaginal smear assessment to euthanize mice at identical timepoints in their estrous cycle. However, this would have been very time consuming and not possible given the large number of mice included in the experiment. As commented by Reviewer 1, we have discussed these limitations in the manuscript (page 28, lines 555-556 and 564-566). For additional emphasis on the need for hormonal analyses, the last sentence of the abstract (page 2-3, lines 54-55) and the last paragraph of the discussion (page 31, lines 621-623) has been modified. 

Other studies using the same mixture of POPs have shown effects on stress response, learning, memory, and altered gene expression patterns of hippocampal genes involved in cognitive function and neuroinflammation in female hybrid 129:C57BL/6 mice and their maternally exposed offspring [8, 10]. Additionally, testicular development, sperm production and sperm chromatin integrity were affected in maternally exposed male hybrid 129:C57BL/6 mice [9]. The mixture of POPs has also been shown to affect intestinal tumorigenesis in a susceptible animal model [6, 7], and alter blood lipid profiles in non-obese diabetic mice [11].

In addition to the mixture of POPs used in the present manuscript, Berntsen and colleagues [1] also created a mixture of POPs, as well as several sub-mixtures, for use in in vitro studies. This mixture was based on chemical concentrations in human blood found in the general Scandinavia population. This in vitro POP mixture has shown effects on nerve and immune cells [12-15], as well as inducing hyperactivity in zebrafish larvae [16, 17]. The total POP mixture or sub-mixtures have also been shown to act antagonistically on the aryl hydrocarbon receptor (AhR) in three different transgenic cell lines [18], and to antagonize transactivation and translocation of the androgen receptor in transfected androgen receptor cell lines [19], to affect in vitro secretion of glucagon-like peptide 1 (GLP-1) in an GLP-1 secreting enteroendocrine cell line [20] and to disrupt steroidogenesis at higher concentrations in H295R cells [21].

In the present manuscript, the conclusions have been moderated throughout the discussion. As commented by Reviewer 1, the design of the study is good and in accordance with OECD guidelines. It included a relatively high number of mice within each exposure group and sampling time. Furthermore, the conclusions have been drawn based on results from conservative statistical methods and, thus, the authors believe that the results presented are not based on speculations. However, as pointed out by both Reviewers, it is important to properly discuss the limitations of the study and we hope that the Reviewer is satisfied with the modifications implemented. 

References

1. Berntsen HF, Berg V, Thomsen C, Ropstad E, Zimmer KE. The design of an environmentally relevant mixture of persistent organic pollutants for use in in vivo and in vitro studies. J Toxicol Env Heal A. 2017;80:1002-16. doi: 10.1080/15287394.2017.1354439.

2. Heeb NV, Schweizer WB, Kohler M, Gerecke AC. Structure elucidation of hexabromocyclododecanes—a class of compounds with a complex stereochemistry. Chemosphere. 2005;61:65-73. doi: 10.1016/j.chemosphere.2005.03.015.

3. Szabo DT, Diliberto JJ, Hakk H, Huwe JK, Birnbaum LS. Toxicokinetics of the flame retardant hexabromocyclododecane gamma: effect of dose, timing, route, repeated exposure, and metabolism. Toxicol Sci. 2010;117:282-93. doi: 10.1093/toxsci/kfq183.

4. Diamanti-Kandarakis E, Bourguignon J-P, Giudice LC, Hauser R, Prins GS, Soto AM, et al. Endocrine-disrupting chemicals: an endocrine society scientific statement. Endocr Rev. 2009;30:293-342. doi: 10.1210/er.2009-0002.

5. Gore AC, Chappell VA, Fenton SE, Flaws JA, Nadal A, Prins GS, et al. EDC-2: the endocrine society's second scientific statement on endocrine-disrupting chemicals. Endocr Rev. 2015;36:E1-e150. doi: 10.1210/er.2015-1010.

6. Hansen KEA, Johanson SM, Steppeler C, Sodring M, Ostby GC, Berntsen HF, et al. A mixture of persistent organic pollutants (POPs) and azoxymethane (AOM) show potential synergistic effects on intestinal tumorigenesis in the A/J Min/+ mouse model. Chemosphere. 2019;214:534-42. doi: 10.1016/j.chemosphere.2018.09.126.

7. Johanson SM, Swann JR, Umu OCO, Aleksandersen M, Muller MHB, Berntsen HF, et al. Maternal exposure to a human relevant mixture of persistent organic pollutants reduces colorectal carcinogenesis in A/J Min/+ mice. Chemosphere. 2020;252:126484. doi: 10.1016/j.chemosphere.2020.126484.

8. Hudecova AM, Hansen KEA, Mandal S, Berntsen HF, Khezri A, Bale TL, et al. A human exposure based mixture of persistent organic pollutants affects the stress response in female mice and their offspring. Chemosphere. 2018;197:585-93. doi: 10.1016/j.chemosphere.2018.01.085.

9. Khezri A, Lindeman B, Krogenæs AK, Berntsen HF, Zimmer KE, Ropstad E. Maternal exposure to a mixture of persistent organic pollutants (POPs) affects testis histology, epididymal sperm count and induces sperm DNA fragmentation in mice. Toxicol Appl Pharmacol. 2017;329:301-8. doi: 10.1016/j.taap.2017.06.019.

10. Myhre O, Zimmer KE, Hudecova AM, Hansen KEA, Khezri A, Berntsen HF, et al. Maternal exposure to a human based mixture of persistent organic pollutants (POPs) affect gene expression related to brain function in mice offspring hippocampus. Under revision. 2021.

11. McGlinchey A, Sinioja T, Lamichhane S, Sen P, Bodin J, Siljander H, et al. Prenatal exposure to perfluoroalkyl substances modulates neonatal serum phospholipids, increasing risk of type 1 diabetes. Environ Int. 2020;143:105935. doi: 10.1016/j.envint.2020.105935.

12. Berntsen HF, Duale N, Bjørklund CG, Rangel-Huerta OD, Dyrberg K, Hofer T, et al. Effects of a human-based mixture of persistent organic pollutants on the in vivo exposed cerebellum and cerebellar neuronal cultures exposed in vitro. Environ Int. 2021;146:106240. doi: 10.1016/j.envint.2020.106240.

13. Berntsen HF, Bølling AK, Bjørklund CG, Zimmer K, Ropstad E, Zienolddiny S, et al. Decreased macrophage phagocytic function due to xenobiotic exposures in vitro, difference in sensitivity between various macrophage models. Food Chem Toxicol. 2018;112:86-96. doi: 10.1016/j.fct.2017.12.024.

14. Davidsen N, Lauvås AJ, Myhre O, Ropstad E, Carpi D, Gyves EM-d, et al. Exposure to human relevant mixtures of halogenated persistent organic pollutants (POPs) alters neurodevelopmental processes in human neural stem cells undergoing differentiation. Reprod Toxicol. 2021;100:17-34. doi: 10.1016/j.reprotox.2020.12.013.

15. Yadav A, Amber M, Zosen D, Labba NA, Huiberts EHW, Samulin Erdem J, et al. A human relevant mixture of persistent organic pollutants (POPs) and perfluorooctane sulfonic acid (PFOS) enhance nerve growth factor (NGF)-induced neurite outgrowth in PC12 cells. Toxicol Lett. 2021;338:85-96. doi: 10.1016/j.toxlet.2020.12.007.

16. Khezri A, Fraser TW, Nourizadeh-Lillabadi R, Kamstra JH, Berg V, Zimmer KE, et al. A mixture of persistent organic pollutants and perfluorooctanesulfonic acid induces similar behavioural responses, but different gene expression profiles in zebrafish larvae. Int J Mol Sci. 2017;18. doi: 10.3390/ijms18020291.

17. Christou M, Fraser TWK, Berg V, Ropstad E, Kamstra JH. Calcium signaling as a possible mechanism behind increased locomotor response in zebrafish larvae exposed to a human relevant persistent organic pollutant mixture or PFOS. Environ Res. 2020;187:109702. doi: 10.1016/j.envres.2020.109702.

18. Doan TQ, Berntsen HF, Verhaegen S, Ropstad E, Connolly L, Igout A, et al. A mixture of persistent organic pollutants relevant for human exposure inhibits the transactivation activity of the aryl hydrocarbon receptor in vitro. Environ Pollut. 2019;254:113098. doi: 10.1016/j.envpol.2019.113098.

19. McComb J, Mills IG, Muller M, Berntsen HF, Zimmer KE, Ropstad E, et al. Human blood-based exposure levels of persistent organic pollutant (POP) mixtures antagonise androgen receptor transactivation and translocation. Environ Int. 2019;132:105083. doi: 10.1016/j.envint.2019.105083.

20. Shannon M, Xie Y, Verhaegen S, Wilson J, Berntsen HF, Zimmer KE, et al. A human relevant defined mixture of persistent organic pollutants (POPs) affects in vitro secretion of glucagon-like peptide 1 (GLP-1), but does not affect translocation of its receptor. Toxicol Sci. 2019;172:359-67. doi: 10.1093/toxsci/kfz192.

21. Ahmed KEM, Frøysa HG, Karlsen OA, Blaser N, Zimmer KE, Berntsen HF, et al. Effects of defined mixtures of POPs and endocrine disruptors on the steroid metabolome of the human H295R adrenocortical cell line. Chemosphere. 2019;218:328-39. doi: 10.1016/j.chemosphere.2018.11.057.

---

## [Decision Letter · Decision Letter 1]

26 May 2021

Perinatal exposure to a human relevant mixture of persistent organic pollutants: effects on mammary gland development, ovarian folliculogenesis and liver in CD-1 mice

PONE-D-20-37867R1

Dear Dr. Johanson,

We’re pleased to inform you that your manuscript has been judged scientifically suitable for publication and will be formally accepted for publication once it meets all outstanding technical requirements.

Kind regards,

Jean-Marc A Lobaccaro, PhD

Academic Editor

PLOS ONE

Additional Editor Comments (optional):

Reviewers' comments:

Reviewer's Responses to Questions

**Comments to the Author**

1. If the authors have adequately addressed your comments raised in a previous round of review and you feel that this manuscript is now acceptable for publication, you may indicate that here to bypass the “Comments to the Author” section, enter your conflict of interest statement in the “Confidential to Editor” section, and submit your "Accept" recommendation.

Reviewer #1: All comments have been addressed

2. Is the manuscript technically sound, and do the data support the conclusions?

Reviewer #1: (No Response)

3. Has the statistical analysis been performed appropriately and rigorously? 

Reviewer #1: (No Response)

4. Have the authors made all data underlying the findings in their manuscript fully available?

Reviewer #1: (No Response)

5. Is the manuscript presented in an intelligible fashion and written in standard English?

Reviewer #1: (No Response)

6. Review Comments to the Author

Reviewer #1: (No Response)

7. PLOS authors have the option to publish the peer review history of their article (what does this mean?). If published, this will include your full peer review and any attached files.

Reviewer #1: **Yes: **Nicolas CHEVALIER

---

## [Editor Report · Acceptance letter]

2 Jun 2021

PONE-D-20-37867R1 

Perinatal exposure to a human relevant mixture of persistent organic pollutants: Effects on mammary gland development, ovarian folliculogenesis and liver in CD-1 mice. 

Dear Dr. Johanson:

I'm pleased to inform you that your manuscript has been deemed suitable for publication in PLOS ONE. Congratulations! Your manuscript is now with our production department. 

Kind regards, 

on behalf of

Dr. Jean-Marc A Lobaccaro 

Academic Editor

PLOS ONE